# In Situ Construction of CNT/CuS Hybrids and Their Application in Photodegradation for Removing Organic Dyes

**DOI:** 10.3390/nano10010178

**Published:** 2020-01-20

**Authors:** Yanping Wang, Fuchuan Jiang, Jiafu Chen, Xiaofeng Sun, Tao Xian, Hua Yang

**Affiliations:** 1School of Science, Lanzhou University of Technology, Lanzhou 730050, China; wangyanpinglut@163.com (Y.W.); jiangfc_sir88@163.com (F.J.); 2Ministry of Education Key Laboratory of Testing Technology for Manufacturing Process, Southwest University of Science and Technology, Mianyang 621010, China; flyingbanana@yeah.net; 3College of Physics and Electronic Information Engineering, Qinghai Normal University, Xining 810008, China; sunxf027@126.com (X.S.); 2016105@qhnu.edu.cn (T.X.)

**Keywords:** CuS nanoparticles/nanoflakes, morphology tailoring, CNT/CuS composites, photodegradation performance

## Abstract

Herein, a coprecipitation method used to synthesize CuS nanostructures is reported. By varying the reaction time and temperature, the evolution of the CuS morphology between nanoparticles and nanoflakes was investigated. It was found that CuS easily crystallizes into sphere-/ellipsoid-like nanoparticles within a short reaction time (0.5 h) or at a high reaction temperature (120 °C), whereas CuS nanoflakes are readily formed at a low reaction temperature (20 °C) for a long time (12 h). Photodegradation experiments demonstrate that CuS nanoflakes exhibit a higher photodegradation performance than CuS nanoparticles for removing rhodamine B (RhB) from aqueous solution under simulated sunlight irradiation. Carbon nanotubes (CNTs) were further used to modify the photodegradation performance of a CuS photocatalyst. To achieve this aim, CNTs and CuS were integrated to form CNT/CuS hybrid composites via an in situ coprecipitation method. In the in situ constructed CNT/CuS composites, CuS is preferably formed as nanoparticles, but cannot be crystallized into nanoflakes. Compared to bare CuS, the CNT/CuS composites manifest an obviously enhanced photodegradation of RhB; notably, the 3% CNT/CuS composite with CNT content of 3% showed the highest photodegradation performance (*η* = 89.4% for 120 min reaction, *k*_app_ = 0.01782 min^−1^). To make a comparison, CuS nanoflakes and CNTs were mechanically mixed in absolute alcohol and then dried to obtain the 3% CNT/CuS-MD composite. It was observed that the 3% CNT/CuS-MD composite exhibited a slightly higher photodegradation performance (*η* = 92.4%, *k*_app_ = 0.0208 min^−1^) than the 3% CNT/CuS composite, which may be attributed to the fact that CuS maintains the morphology of nanoflakes in the 3% CNT/CuS-MD composite. The underlying enhanced photocatalytic mechanism of the CNT/CuS composites was systematically investigated and discussed.

## 1. Introduction

Water resources, being the indispensable foundation for the survival of human beings, are getting seriously polluted due to rapid industrialization. Wastewater largely produced from chemical industries carries a large number of harmful pollutants. As the dominant pollutants in the industrial wastewater, organic dyes, such as rhodamine B (RhB), have carcinogenic properties and seriously threaten the health of mankind and aquatic life. Various technologies have been developed to treat industrial wastewater, among which semiconductor-based photocatalysis has received much interest as a “green” technology in wastewater purification by using solar radiation as the power source [1,2,3,4]. When semiconductor photocatalysts are irradiated with sunlight, valence band (VB) holes (h^+^) and conduction band (CB) electrons (e^−^) are produced through a process of photoexcitation. The photoproduced holes and electrons are the basic reactive species, which directly or indirectly participate in oxidation and/or reduction reactions to cause the decomposition of organic dyes into harmless inorganic molecules. However, the photoactivity of semiconductors are generally limited due to the ease of recombination of the photoexcited holes and electrons. Many avenues have therefore been adopted to modify photocatalysts with the aim of facilitating the spatial separation of the photogenerated carriers and ensuring their increased utilization for photodegradation reactions [5,6,7,8,9,10].

Carbon and noble metal nanomaterials have become a research hotspot owing to their intriguing physicochemical properties and great potential applications in biomedicine, electronic devices, sensors and wave absorption, etc., [11,12,13,14,15,16,17,18,19]. These interesting nanomaterials have also been demonstrated to be excellent modifiers that can be used to improve the photodegradation performances of semiconductor photocatalysts [20,21,22,23]. Owing to their good electron trapping properties, carbon/noble metal nanomaterials can be used to trap photogenerated electrons in photocatalysts, thus efficiently preventing their recombination with photogenerated holes. Furthermore, nanosized noble metals also exhibit interesting localized surface plasmon resonance (LSPR) properties, and nanocarbons exhibit interesting up-conversion photoluminescence (PL) properties [24,25]. Due to these interesting properties, noble metal/carbon decorated photocatalysts can enhance the absorption and utilization of visible light during the photodegradation process. The construction of heterojunctions between two semiconductors is another important avenue to facilitate the photogenerated electron/hole pair separation benefiting from the carrier transfer between two semiconductors [26,27,28].

Copper oxides and sulfides have attracted a great deal of interest due to their excellent optical/electronic properties and great application prospects in many fields (e.g., solar cells, optical filters, lithium-ion batteries, supercapacitor, sensors, and catalysts) [29,30,31,32]. Copper sulfides have several variants, from copper-deficient CuS_2_ to copper-rich Cu_2_S. Among them, covellite (CuS) has been the most intensively studied. Compared to the famous semiconductor photocatalysts TiO_2_ and ZnO [33,34], CuS has a relatively narrow bandgap energy, i.e., ~2.0 eV. This suggests that CuS can efficiently harvest the visible light of the solar spectrum, and is particularly interesting in the application as a visible-light-driven photocatalyst [35,36]. However, the photoactivity of bare CuS is not sufficient, due to the ease of recombination of the photoexcited holes and electrons. To overcome this shortcoming, some strategies have been developed, including the construction of heterojunctions with other semiconductors, and decoration with graphene/carbon quantum dots/noble metal nanoparticles [37,38,39,40]. Several research groups have integrated CuS with carbon nanotubes (CNTs), and demonstrated the promising application of the derived CNT/CuS composites in solar cells, semiconductor plasmonics, and supercapacitors [41,42,43,44,45,46,47,48]. However, little work has been undertaken on the photocatalytic application of CNT/CuS hybrid composites for the degradation of organic dyes.

Herein we adopted a coprecipitation method to synthesize CuS nanoparticles and nanoflakes. The evolution process of CuS morphology between nanoparticles and nanoflakes was systematically investigated by varying the reaction time and temperature. Furthermore, we immobilized CuS nanoparticles on CNTs via a in situ coprecipitation method to construct CNT/CuS hybrid composite photocatalysts. By using rhodamine B (RhB) (dissolved in water) as the model pollutant and simulated sunlight as the light source, the photodecomposition performances of the as-prepared CuS and CNT/CuS composite photocatalysts were investigated. The underlying photodegradation mechanism of the CNT/CuS composites was systematically investigated and discussed.

## 2. Materials and Methods

### 2.1. Synthesis of CuS Nanoparticles/Nanoflakes

All chemical reagents, being of analytical grade, were purchased from chemical reagent corporations in China (Shanghai Aladdin Reagent Co., Ltd., Shanghai, China) and used directly in the present experiments. The synthesis of CuS nanoflakes was based on a coprecipitation method. In a typical synthesis process, 0.4832 g (2 mmol) of Cu(NO_3_)_2_∙3H_2_O was dissolved in 30 mL ethylene glycol with 30 min of magnetic stirring (designated as solution A). Then, 0.4802 g (2 mmol) of Na_2_S∙9H_2_O was dissolved in 20 mL ethylene glycol with 30 min of magnetic stirring (designated as solution B). Solution B was then slowly dropped into solution A. The obtained mixture solution was reacted for a certain time (0.5, 2, 10 or 12 h) at a selected temperature (20, 70 or 120 °C) under magnetic stirring. The produced precipitate was collected by centrifugation, rinsed with deionized water (three time)/ethanol (three time), and dried at 60 °C for 12 h. The final product was obtained as CuS nanoparticles or nanoflakes. The sample prepared at time *t* h and temperature *T* °C was termed *t* h-*T* °C.

### 2.2. Preparation of CNT/CuS Composites

CNT/CuS composites were constructed in situ by the coprecipitation method, as described above. During the coprecipitation synthesis process of CuS, a stoichiometric amount of CNTs was dispersed in the precursor Cu(NO_3_)_2_ solution, to which was then dropped slowly with Na_2_S solution. The coprecipitation reaction temperature was 20 °C and the reaction time was 12 h. By loading different amounts of CNTs in the precursor solution, several composite samples (1%CNT/CuS, 3% CNT/CuS and 5%CNT/CuS) were prepared. For comparison, a mechanical mixing-drying method was also used to prepare CNT/CuS composites. First, 0.01 g of CNTs and 0.32 g of CuS nanoflakes (prepared at 20 °C for 12 h, i.e., the 12 h-20 °C CuS sample) were uniformly mixed in absolute alcohol and then dried at 60 °C. The derived composite was designated as 3% CNT/CuS-MD.

### 2.3. Sample Characterization Methods

A D8 Advance X-ray diffractometer (Bruker AXS, Karlsruhe, Germany) with λ_Cu-kα_ = 0.15406 nm was used to record the X-ray powder diffraction (XRD) patterns of the samples. Scanning/transmission electron microscopy (SEM/TEM) investigations were performed on a JSM-6701F field-emission scanning electron microscope (JEOL Ltd., Tokyo, Japan) and a JEM-1200EX field-emission transmission electron microscope (JEOL Ltd., Tokyo, Japan). A PHI-5702 multi-functional X-ray photoelectron spectrometer (Physical Electronics, hanhassen, MN, USA) was used for the X-ray photoelectron spectroscopy (XPS) analyses. Fourier transform infrared (FTIR) spectroscopy was analyzed on a Spectrum Two FTIR spectrophotometer (PerkinElmer, Waltham, MA, USA).

The photoelectrochemical properties of the samples, i.e., electrochemical impedance spectroscopy (EIS) and photocurrent response, were measured on a CST 350 electrochemical workstation according to the procedure elaborated in our previous work [47]. A Na_2_SO_4_ aqueous solution (0.1 mol L^−1^) served as the electrolyte. Simulated sunlight emitted from a 200-W xenon lamp (300 < λ < 2500 nm) was used as the light source. The preparation process of the working electrode was described in our previous work [49].

### 2.4. Photocatalytic Testing Procedure

RhB in aqueous solution (5 mg L^−1^) was used as the model pollutant to evaluate the photodegradation performances of the prepared photocatalysts under illumination of simulated sunlight emitting from a 200-W xenon lamp. First, 0.03 g of the photocatalyst and 100 mL of RhB solution were loaded in the photocatalytic reactor with a capacity of 200 mL. Prior to photocatalysis, the mixture solution was placed in the dark under magnetic stirring for 30 min to determine the adsorption of RhB onto the photocatalyst. During the photodegradation process, the residual RhB concentration was monitored by sampling 2.5 mL of the reaction solution and measuring its absorbance on a UV-vis spectrophotometer at λ_RhB_ = 554 nm. The photocatalyst was centrifugally removed from the reaction solution before measuring its absorbance. The degradation percentage (*η*) of RhB after photoreaction for *t* min was defined as *η* = (*C*_0_ − *C*_t_)/*C*_0_ × 100% (*C*_0_ = initial RhB concentration, *C*_t_ = residual RhB concentration at *t* min).

## 3. Results and Discussion

### 3.1. Properties of Bare CuS Photocatalysts

The morphologies of the CuS samples synthesized at 20 °C with different reaction times (0.5, 2, 10, and 12 h) were observed by SEM, as shown in Figure 1 and Appendix A. It may be seen that, at a short reaction time, i.e., 0.5 h, CuS is crystallized into sphere-/ellipsoid-like nanoparticles of 10–25 nm in size (Figure 1a). By increasing the reaction time, the CuS nanoparticles gradually grew along the directions with higher surface energies, and small-sized nanoflakes were obtained at 2 h. When the reaction time increased to 12 h, the nanoparticles almost disappeared, and pure CuS nanoflakes were formed (Figure 1d). The CuS nanoflakes, having diameter sizes of 300–500 nm with thicknesses of 20–30 nm, interacted to form nanoflake networks. The evolution of CuS nanoflakes can be explained by the Ostwald ripening mechanism. According to this mechanism, The nucleation process of CuS crystals occurs in the region of supersaturated fluid. By increasing the reaction time, the CuS nuclei gradually grow into nanoflakes due to the anisotropic surface energy in CuS crystals. All the as-prepared CuS samples, including the nanoparticles and nanoflakes, appear black in color, as seen from the digital images inserted in the SEM images of Figure 1, implying that they exhibit strong visible-light absorption [50]. The strong visible-light absorption of CuS samples is further confirmed by the ultraviolet-visible diffuse reflectance spectroscopy (UV-vis DRS) spectra (Appendix A).

Figure 2 shows the SEM images of CuS samples synthesized at different reaction temperatures (20 °C, 70 °C, and 120 °C) for 10 h. It was observed that with increasing the reaction temperature, CuS nanoflakes evolved into nanoparticles. The possible reason for this evolutional behavior is that the anisotropic surface energy of CuS crystals could be significantly influenced by high temperatures, and thus, the preferential growth does not happen; moreover, the increase of nucleation sites for CuS formation at high temperatures may reduce the chance for the anisotropic growth of nanoflakes to occur. The CuS samples obtained at different reaction temperatures also exhibited strong visible-light absorption, as demonstrated by their black color (inserted in the SEM images of Figure 2) and UV-vis DRS spectra (Appendix A).

Figure 3 shows the XRD patterns of CuS samples synthesized at different temperatures with different reaction times. For all the samples, the diffraction peaks can be perfectly indexed according to the standard diffraction data of PDF#06-0464, indicating that all the samples are crystallized into pure CuS hexagonal structures (space group: P63/mmc). 

The photodegradation performances of the as-synthesized CuS samples were evaluated by removing RhB from aqueous solution under irradiation of simulated sunlight. Figure 4a illustrates the adsorption and time-dependent photodegradation curves of RhB over the CuS samples synthesized at 20 °C with different reaction times. A large adsorption of RhB was observed for all the CuS samples (*η* = 35.8–48.1%), which may have been caused by the electrostatic interaction between cationic RhB molecules and −OH groups attached to CuS [51]. Under 120 min irradiation, the degradation percentage of RhB was observed to be *η* = 64.1% for sample 0.5 h-20 °C and *η* = 79.2% for sample 12 h-20 °C. This implies that the derived CuS nanoflakes had higher photodegradation activity than the CuS nanoparticles. It is generally accepted that the morphology of photocatalysts has an important effect on their photocatalytic performances. Figure 4b shows the corresponding degradation kinetic plots, which are modeled using the pseudo-first-order kinetic equation Ln (*C*_t_/*C*_0_) = −*k*_app_*t* [52]. The apparent first-order reaction rate constant *k*_app_ can be used for quantitative comparisons of the photodegradation activity between the samples. Based on the values of *k*_app_, it was concluded that the CuS nanoflakes (sample 12 h-20 °C) manifested a photodegradation activity that was about 1.6 times higher that of CuS nanoparticles (sample 0.5 h-20 °C). Figure 4c,d show the photodegradation curves of RhB and the corresponding degradation kinetic plots over the CuS samples synthesized at different reaction temperatures for 10 h, respectively. By increasing the synthesis temperature, the derived CuS samples exhibited a decrease in photodegradation activity. This could be attributed to the fact that CuS nanoflakes are transformed into nanoparticles at high reaction temperatures.

### 3.2. Properties of CNT/CuS Composite Photocatalysts

It was demonstrated that the CuS sample synthesized at 20 °C for 12 h (i.e., sample 12 h-20 °C) had a morphology of nanoflakes, and exhibited the highest photodegradation activity. Based on this information, we attempted to in situ assemble CuS nanoflakes with CNTs under the same synthesis conditions for sample 12 h-20 °C by loading CNTs in the precursor solution. Figure 5a shows the SEM image of the as-derived hybrid composite 3% CNT/CuS, and the SEM image of pure CNTs is given in Appendix A for comparison. Surprisingly, it was observed that CuS in the composite was crystallized into nanoparticles, instead of forming nanoflakes. The composite was obviously constructed by coupling CuS nanoparticles with CNTs. A possible reason for this phenomenon is that CNTs could greatly influence the anisotropic surface energy of CuS crystals, and block the anisotropic expansion of the crystal seed, and thereby, the growth of CuS nanoflakes. The XRD pattern (Figure 5b) indicates that the crystal structure of CuS in the composite undergoes no change (maintaining hexagonal structure). In addition, the CNT/CuS composites exhibit strong visible-light absorption, as demonstrated by their UV-vis DRS spectra (Appendix A).

TEM investigation was performed to further reveal the microstructure of the 3% CNT/CuS composite. Figure 6a,b display the TEM images of 3% CNT/CuS, clearly showing that CuS nanoparticles are well coupled with CNTs to form CNT/CuS hybrid composite photocatalysts. The selected area electron diffraction (SAED) pattern inserted in Figure 6b shows obvious polycrystalline diffraction rings, which can be perfectly indexed to the hexagonal CuS structure. The high-resolution TEM (HRTEM) image depicted in Figure 6c further confirms the good coupling of CuS nanoparticles with CNTs. Perfect lattice fringes are observed for the CuS nanoparticles, implying that they are well crystallized. Energy-dispersive X-ray spectroscopy (EDS) analysis suggested that the composite was composed of the elements Cu/S/C and minor O (Figure 6d). The appearance of minor O element indicated the possible existence of other groups (e.g., CO_3_^2−^ groups) in the composite. From the dark-field scanning TEM (DF-STEM) image (Figure 6e) and corresponding EDS elemental mapping images (Figure 6f–h), it is clearly seen that the nanoparticles are composed of Cu/S elements. The distribution of elemental C throughout the composite implies that the CNTs were integrated with the CuS nanoparticles.

XPS is a useful technique for the determination of chemical states of elements [53,54]. Figure 7a displays the survey scan XPS spectrum of the 3% CNT/CuS composite, confirming that elemental Cu and S are included in the composite. The minor levels of detected elemental O were indicative of the possible existence of other groups in the composite, such as CO_3_^2−^ groups. The high-resolution core-level XPS spectra of Cu-2p, S-2p and C-1s are depicted in Figure 7b–d, respectively. On the Cu-2p core-level XPS spectrum (Figure 7b), the appearance of two strong peaks at 932.4 (Cu-2p_3/2_) and 952.3 eV (Cu-2p_1/2_) confirmed the presence of Cu^2+^ species in the CuS crystals [55]. S^2−^ species were confirmed by the detected two peaks at 161.9 (S-2p_3/2_) and 163.3 eV (S-2p_1/2_) on the S-2p core-level XPS spectrum (Figure 7c) [56]. The C-1s core-level XPS spectrum (Figure 7d) can be deconvoluted into four peaks at 284.4, 286.1, 287.9, and 290.0 eV, which are characterized as C=C (sp^2^), C−C (sp^3^), C=O and π–π* shake-up feature [57]. This implies the possible existence of defects and other functional groups on the surface of CNTs.

Figure 8 shows the FTIR spectra of CuS nanoflakes (i.e., sample 12 h-20 °C), pure CNTs, and the 3% CNT/CuS composite. On the FTIR spectrum of CuS, the observed strong absorption peak at 630 cm^−1^ (Cu–S stretching mode) indicates the formation of CuS crystals [58]. The infrared vibration peaks detected at 1108 and 1395 cm^−1^ are indicative of the possible presence of CO_3_^2−^ groups [59]. Water molecules were confirmed to be adsorbed on the CuS surface by the peak at 1634 cm^−1^ [60,61,62]. On the FTIR spectrum of CNTs, no obvious vibration peaks of carbon skeleton (C=C) featured in CNTs [63] were detected, possibly due to their weak absorption intensity. The appearance of a C=O vibration peak at 1733 cm^−1^ indicates that CNTs were bound with carboxyl (–COOH) group. For the 3% CNT/CuS composite, the characteristic absorption peak of CuS was observed on its FTIR spectrum, but its intensity was much lower than that of bare CuS. CO_3_^2−^ groups and water molecules could also exist in the CNTs and 3% CNT/CuS composite due to the appearance of absorption peaks at 1108, 1395 and1634 cm^−1^. In addition, several additional peaks were detected on the FTIR spectrum of the 3% CNT/CuS composite compared to bare CNTs and CuS, which implies that other C-based groups could be formed in 3% CNT/CuS during the coprecipitation reaction process.

The Raman spectra recorded from CuS nanoflakes (12 h-20 °C), pure CNTs, and the 3% CNT/CuS composite are shown in Figure 9. A strong peak at 465 cm^−1^ was observed on the Raman spectrum of CuS, corresponding to the A_1g_ vibration mode of CuS crystals [64]. Five peaks (1341, 1587, 2681, 2923 and 3216 cm^−1^) were detected on the Raman spectrum of pure CNTs. The strong peaks appearing at 1341 and 1587 cm^−1^ are characterized as the disorder-induced D mode and in plane vibrational G mode of CNTs, respectively. The three weak peaks at 2681, 2923, and 3216 cm^−1^ arose due to the 2D, D+G and 2D’ Raman bands, respectively [65]. The characteristic Raman peaks of CuS and CNTs were observed for the 3% CNT/CuS composite, implying the coupling of CuS with CNTs.

Photoelectrochemical measurements, including EIS and photocurrent response, are used to elaborate the separation/transfer behavior of photoexcited carriers of CuS nanoflakes (12 h-20 °C) and CNT/CuS composites. As seen in Figure 10a, the EIS Nyquist plots for all samples present a typical semicircle at the high-frequency region, followed by a straight line at the low-frequency region. It is well established that the semicircle diameter can be used as an indicator for the charge-transfer resistance [66,67]. The CNT/CuS composites show a semicircle diameter smaller than that for CuS; notably, the smallest semicircle diameter was observed for 3% CNT/CuS. The transient photocurrent-time response curves, recorded under intermittent irradiation of the photocatalysts with simulated sunlight, are shown in Figure 10b. Upon irradiation, an obvious photocurrent was observed for all the samples, which suddenly dropped to almost zero when the light was turned off. The photocurrent density of the CNT/CuS composites (especially 3% CNT/CuS) was obviously higher than that of bare CuS. The photoelectrochemical measurements suggest that the CNT/CuS composites (particularly 3% CNT/CuS) have a more efficient separation of e^−^/h^+^ pairs and better interface carrier-transfer property than bare CuS.

Figure 11a illustrates the adsorption and time-dependent photodecomposition curves of RhB photocatalyzed by CuS nanoflakes (12 h-20 °C) and CNT/CuS composites, as well as 3% CNT/CuS-MD (obtained by a mechanical mixing-drying method) for comparison. A slightly increased adsorption of RhB was observed for the CNT/CuS composites compared to bare CuS. The possible reason for this is that CuS nanoflakes evolve into nanopartilces in the CNT/CuS composites, which have a relatively larger dye adsorption than the nanoflakes. Moreover, the introduced CNTs could also have a large dye adsorption. It was noted that in the CNT/CuS composites prepared through an in situ coprecipitation method, CuS was crystallized into nanoparticles instead of nanoflakes. Although CuS nanoparticles have a photodegradation activity that is lower than that of CuS nanoflakes (see Figure 5), the derived CNT/CuS composites still manifest an improved photodegradation for RhB compared to bare CuS nanoflakes, as illustrated in Figure 11a. The 3% CNT/CuS composite with CNTs content of 3% exhibited the highest photodegradation performance, reaching *η* = 89.4% with 120 min of simulated-sunlight irradiation. To make a comparison, CuS nanoflakes and CNTs were mechanically mixed in absolute alcohol and then dried to obtain the 3% CNT/CuS-MD composite. It was observed that the 3% CNT/CuS-MD composite exhibited a slightly higher photodegradation performance (*η* = 92.4%) than the 3% CNT/CuS composite. This phenomenon could be due to the fact that CuS maintains a morphology of nanoflakes in 3% CNT/CuS-MD, as seen from the SEM image in Appendix A. By modeling the degradation kinetic plots (Figure 11b) with Ln (*C*_t_/*C*_0_) = −*k*_app_*t*, the values of *k*_app_ (i.e., apparent first-order reaction rate constant) could be derived. According to the *k*_app_ values, the photodegradation activities of 3% CNT/CuS and 3% CNT/CuS-MD were determined to be ~1.5 and ~1.7 times higher that of pure CuS nanoflakes.

To examine the reusability of the 3% CNT/CuS composite photocatalyst for the photocatalytic removal of RhB, it was collected and rinsed with deionized water after the photodegradation experiment. The recovered photocatalyst was dispersed in fresh RhB solution to repeat the photodegradation experiment under the same conditions. Figure 11c illustrates the removal percentage of RhB (photoreaction for 120 min) over 3% CNT/CuS with four times of cycles. Only a minor loss of the dye degradation percentage was found after four cycles of reuse, indicating an excellent stability of the 3% CNT/CuS composite photocatalyst for the photodegradation application.

To determine the active species in the 3% CNT/CuS photocatalytic system, reactive species trapping experiments were carried out. First, 0.1 mmol of methyl alcohol (MeOH, scavenger of photogenerated h^+^), 0.1 mmol of iso-propyl alcohol (IPA, scavenger of hydroxyl •OH), and 0.1 mmol of benzoquinone (BQ, scavenger of superoxide •O_2_^−^) [68] were separately added in the photodegradation system. Then, the photodegradation experiment was performed under the same procedure to that without adding scavengers. Figure 11d shows the effect of the scavengers on the degradation percentage of RhB with 120 min photoreaction. RhB degradation was observed to be obviously inhibited on the addition of BQ and MeOH; in contrast, IPA had a very small effect on the dye degradation. This implies that the removal process of the dye is mainly related to the photogenerated holes and •O_2_^−^ radicals, whereas •OH plays a minor role in the dye degradation.

To determine the energy bandgap (*E*_g_) of CuS, Figure 12a shows the Tauc plot of (*αhν*)^2^ vs. *hν* derived from the UV-vis DRS spectrum of CuS (0.5 h-20 °C) by using the Tauc relation [69], where *α* is the Kubelka-Munk (K-M) absorption coefficient and *hν* is the photon energy. By extrapolating the linear portion of the Tauc plot to the *x*-axis, the *E*_g_ value of CuS was derived as 2.02 eV. The CB and VB potentials of CuS can be estimated using the Mott-Schottky (M-S) method [70]. Figure 12b shows the M-S plot of CuS derived on the base of the electrochemical measurement at a given frequency of 5000 Hz. By extrapolating the linear portion of the M-S plot to the *x*-axis, the flat band potential (*V*_FB_) of CuS, approximately equal to its VB potential due to its p-type semiconductivity (negative slope of the M-S plot), was estimated to be +0.86 V vs. standard calomel electrode (SCE), correspondingly +1.52 V with reference to normal hydrogen electrode (NHE) (*V*_NHE_ = *V*_SCE_ + 0.059 × pH (=7) + 0.242 [70]). The CB potential of CuS was therefore obtained as −0.50 V vs. NHE according to *E*_g_ = *E*_CB_ − *E*_VB_.

Figure 12c schematically depicts the photodegradation mechanism of the CNT/CuS composite photocatalysts. When the CNT/CuS composites are irradiated with simulated sunlight, both CuS and CNT are photoexcited. CuS is photoexcited to produce holes in its VB and electrons in its CB. CNT is photoexcited to produce electrons in its lowest unoccupied molecular orbital (LUMO) and holes in its highest occupied molecular orbital (HOMO) [71]. It is well known that CNT is an excellent electron acceptor, which will trap the CB electrons of CuS. Simultaneously, the photoexcited CNT is a good electron donor, readily leading to the migration of the photoexcited electrons in the LUMO of CNT to the CB of CuS. The interesting electron transfer process is somewhat similar to the Z-scheme charge mechanism (but different from the type II heterojunction [72]), and can efficiently prevent the recombination of the CB electrons with the VB holes in CuS. This is the dominant mechanism resulting in the enhanced photodegradation performance of the CNT/CuS composites compared to bare CuS. Furthermore, the photoexcited electrons in the LUMO of CNT could also participate in the photodegradaton reactions. It was noted that the CB potential of CuS (−0.50 V vs. NHE) was more negative than the redox potential of O_2_/•O_2_^−^ (−0.13 V vs. NHE) [73]. This implies that •O_2_^−^ radicals, which are determined to be an important type of reactive species in the photodegradation reactions, can be easily produced through the reactions of O_2_ species with the CB electrons of CuS, as well as the LUMO electrons of CNT. The VB holes of CuS (+1.52 V vs. NHE) are not sufficiently positive to react with OH^−^ or H_2_O species to produce •OH radicals, due to the high redox potentials of OH^–^/•OH (+1.99 V vs. NHE) and H_2_O/•OH (+2.38 V vs. NHE) [74]. As a result, the direct oxidation by the VB holes of CuS becomes the dominant mechanism causing the dye decomposition, as elucidated by the reactive species trapping experiments. Under the action of the reactive species •O_2_^−^ and h^+^, RhB is decomposed into harmless inorganic molecules such as H_2_O, CO_2_, and NH_4_^+^. The photodegradation mechanism can be briefly described as follows.

CNT/CuS + *hν* → CNT(*e*_LUMO_^−^ + *h*_HOMO_^+^)/CuS(*e*_CB_^−^ + *h*_VB_^+^) (1)CNT(*e*_LUMO_^−^ + *h*_HOMO_^+^)/CuS(*e*_CB_^−^ + *h*_VB_^+^) → CNT(*e*_LUMO_^−^)/CuS(*h*_VB_^+^) (2)CNT(*e*_LUMO_^−^) + O_2_ → •O_2_^−^ (3)•O_2_^−^, CuS(*h*_VB_^+^) + RhB → H_2_O, CO_2_, NH_4_^+^ (4)

## 4. Conclusions

We have adopted a coprecipitation approach to synthesize CuS nanostructures, and found that their morphology is highly dependent on the reaction time and temperature. With a short reaction time (0.5 h) or a high reaction temperature (120 °C), CuS nanoparticles of 10–25 nm in size are formed, whereas at a low reaction temperature (20 °C) for a long time (12 h), the synthesis of CuS nanoflakes with diameter of 300–500 nm and thickness of 20–30 nm occurs. The derived CuS nanoflakes exhibit a higher photocatalytic removal of RhB under simulated sunlight irradiation than CuS nanoparticles. CNT/CuS hybrid composites were constructed via an in situ coprecipitation method. Owing to the great effect of CNTs on the anisotropic surface energy of CuS crystals, CuS nanoflakes cannot be formed in the in situ constructed CNT/CuS composites. Instead, CuS nanoparticles are preferably formed. An obviously enhanced photodegradation performance is observed for the CNT/CuS composites, particularly the optimal composite sample—3% CNT/CuS manifests a photodegradation performance ~1.5 times as large as that of bare CuS nanoflakes. The 3% CNT/CuS-MD composite obtained by mechanically mixing CuS nanoflakes and CNTs in absolute alcohol and then drying exhibited a photodegradation performance which was slightly higher than that of the 3% CNT/CuS composite, due to the fact that the morphology of CuS nanoflakes in the 3% CNT/CuS-MD composite underwent no change. The enhanced photocatalytic mechanism of the CNT/CuS composites was mainly ascribed to the electron transfer process between CNT and CuS, which efficiently prevented the electron/hole pair recombination in CuS.

## Figures and Tables

**Figure 1 nanomaterials-10-00178-f001:**
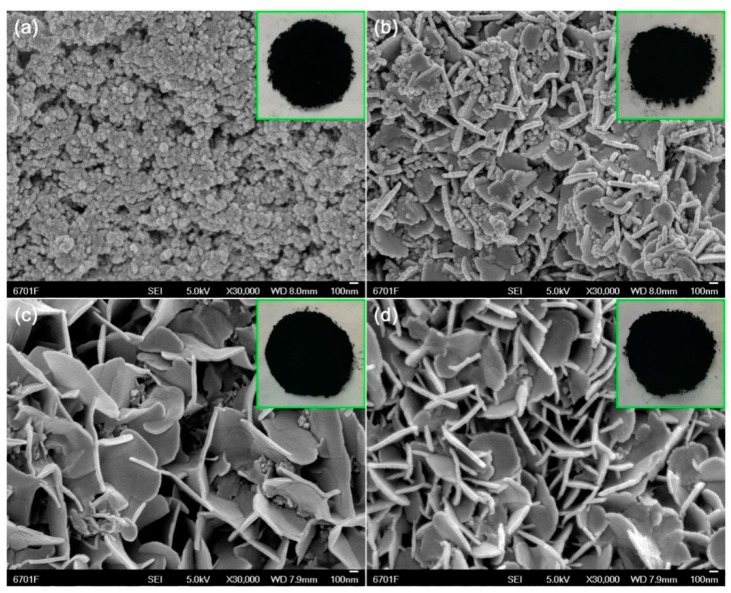
Scanning electron microscopy (SEM) images of CuS samples synthesized at 20 °C with different reaction times. (**a**) 0.5 h-20 °C, (**b**) 2 h-20 °C, (**c**) 10 h-20 °C, (**d**) 12 h-20 °C.

**Figure 2 nanomaterials-10-00178-f002:**
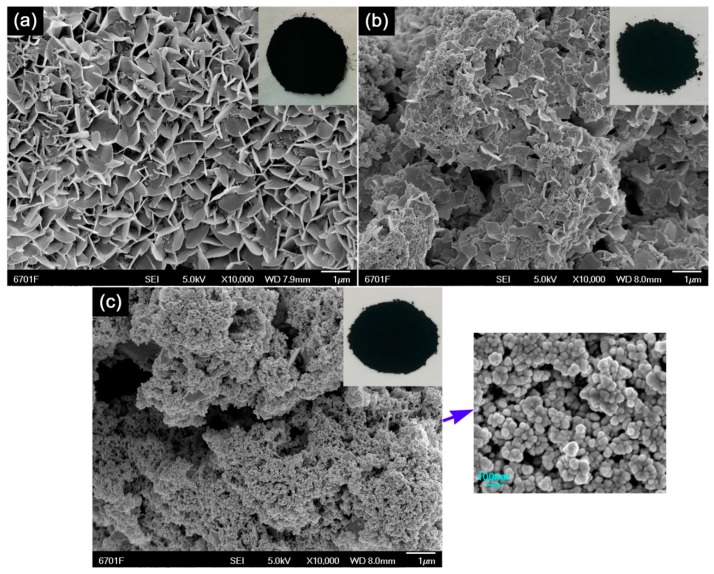
SEM images of CuS samples synthesized at different reaction temperatures for 10 h. (**a**) 10 h-20 °C, (**b**) 10 h-70 °C, (**c**) 10 h-120 °C.

**Figure 3 nanomaterials-10-00178-f003:**
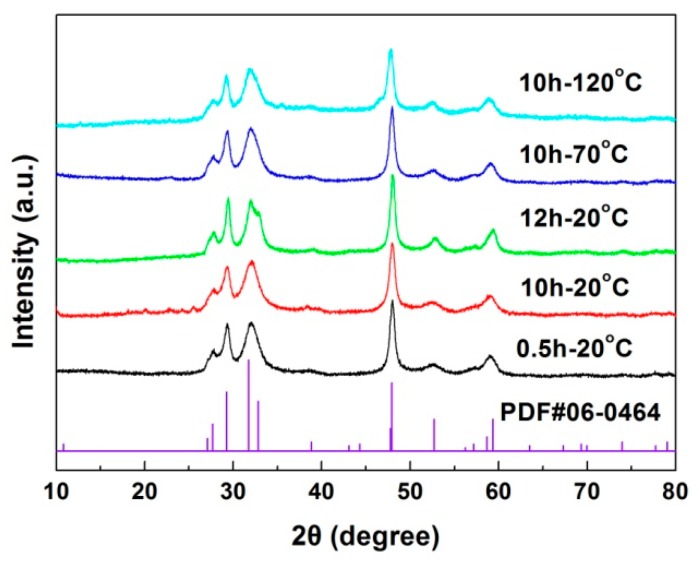
X-ray powder diffraction (XRD) patterns of CuS samples synthesized at different temperatures with different reaction times.

**Figure 4 nanomaterials-10-00178-f004:**
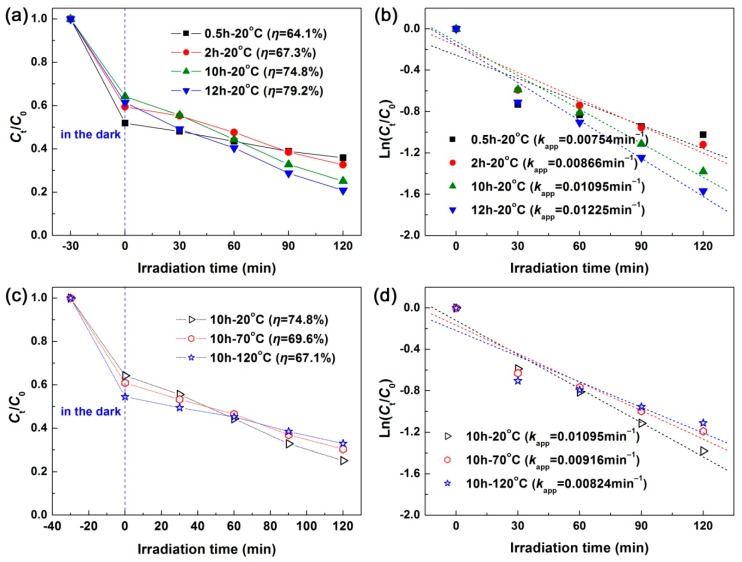
(**a**,**b**) Photodegradation curves of RhB and corresponding degradation kinetic plots over the CuS samples synthesized at 20 °C with different reaction times, respectively. (**c**,**d**) Photodegradation curves of RhB and corresponding degradation kinetic plots over the CuS samples synthesized at different reaction temperatures for 10 h, respectively.

**Figure 5 nanomaterials-10-00178-f005:**
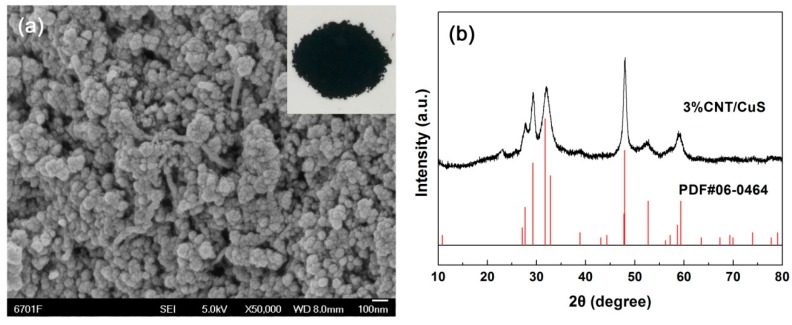
SEM image (**a**) and XRD pattern (**b**) of the 3% CNT/CuS composite.

**Figure 6 nanomaterials-10-00178-f006:**
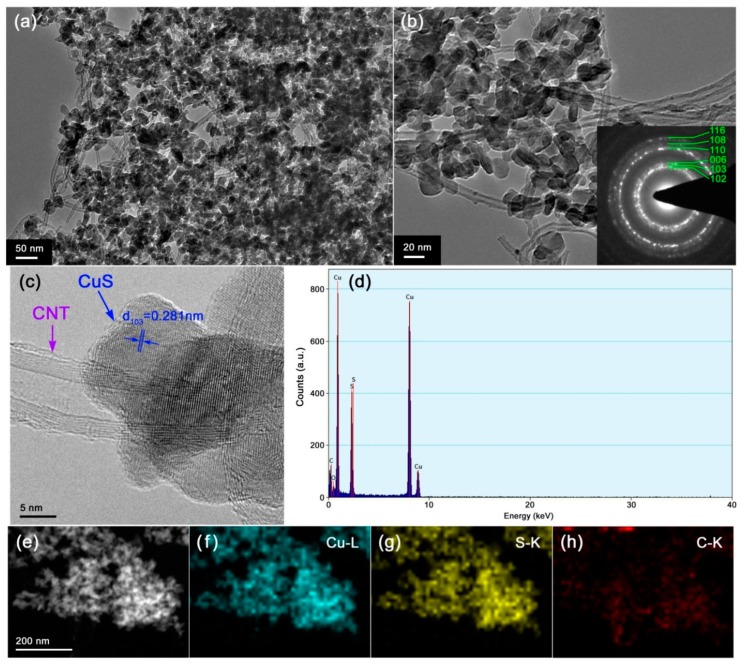
Transmission electron microscopy (TEM) images (**a**,**b**), selected area electron diffraction (SAED) pattern (inset in (**b**)), high-resolution TEM (HRTEM) image (**c**), Energy-dispersive X-ray spectroscopy (EDS) spectrum (**d**), dark-field scanning TEM (DF-STEM) image (**e**), and EDS elemental mapping images (**f**–**h**) of the 3% CNT/CuS composite.

**Figure 7 nanomaterials-10-00178-f007:**
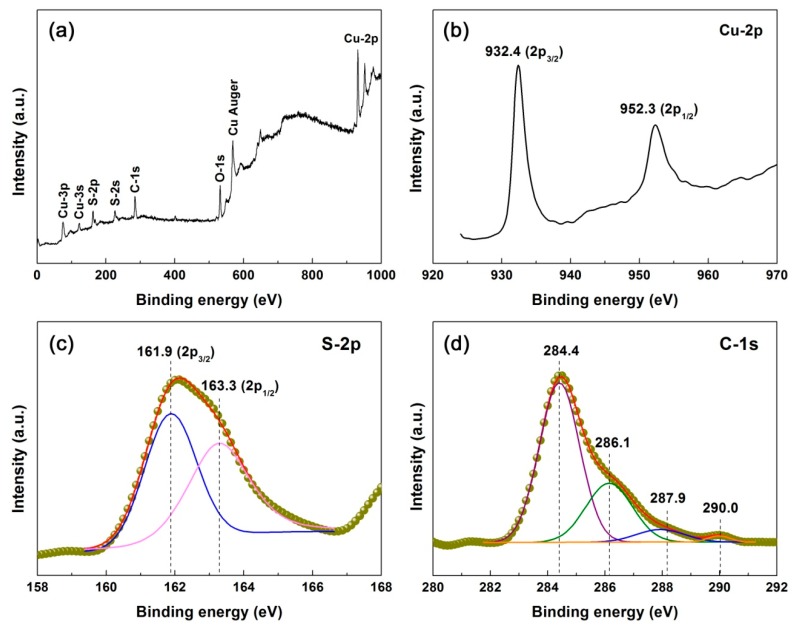
Survey scan XPS spectrum (**a**) and high-resolution XPS spectra of (**b**) Cu-2p, (**c**) S-2p, and (**d**) C-1s of the 3% CNT/CuS composite.

**Figure 8 nanomaterials-10-00178-f008:**
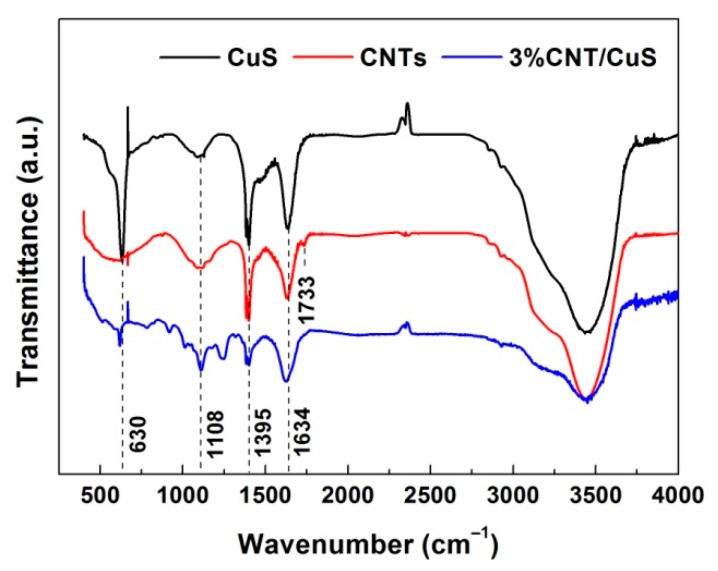
FTIR spectra of CuS (12 h-20 °C), CNTs and 3% CNT/CuS.

**Figure 9 nanomaterials-10-00178-f009:**
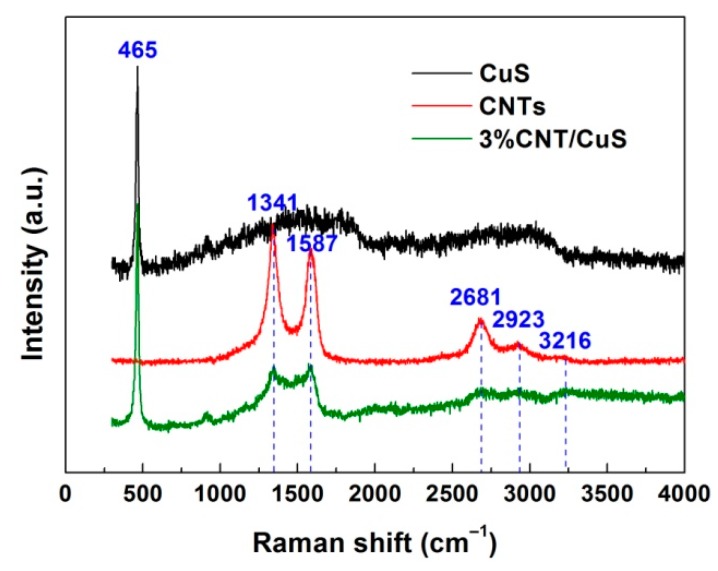
Raman spectra of CuS (12 h-20 °C), CNTs, and 3% CNT/CuS.

**Figure 10 nanomaterials-10-00178-f010:**
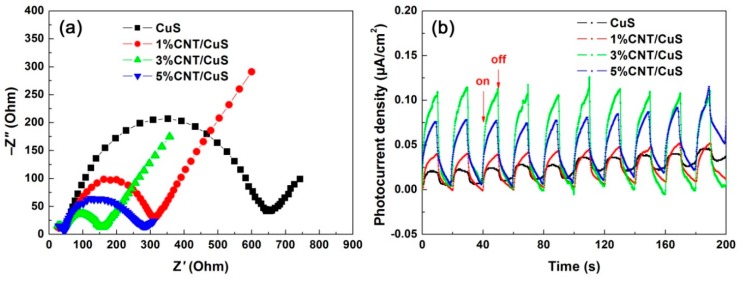
Nyquist plots of the EIS spectra (**a**) and transient photocurrent-time response curves (**b**) of CuS (12 h-20 °C) and CNT/CuS composites.

**Figure 11 nanomaterials-10-00178-f011:**
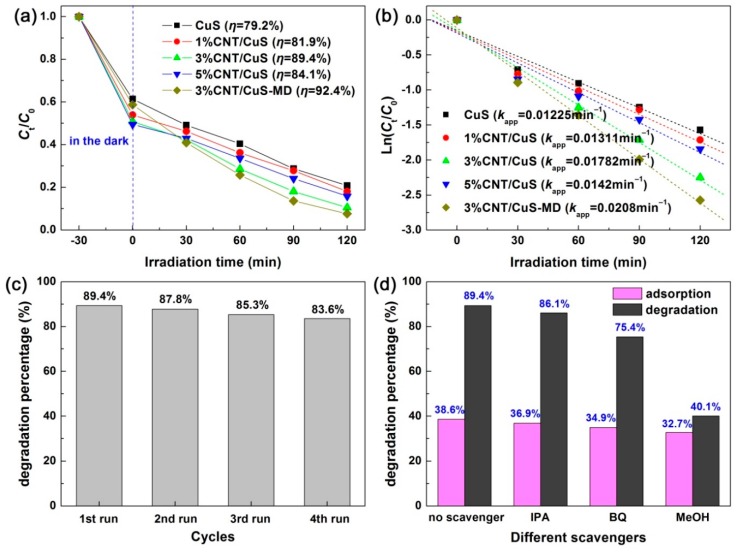
(**a**) Time-dependent photodegradation curves of RhB over CuS (12 h-20 °C) and CNT/CuS composites. (**b**) Kinetic plots of the RhB photodegradation. (**c**) Photodegradation percentage of RhB over 3% CNT/CuS repeatedly used for four times. (**d**) Effects of IPA, BQ, and MeOH on the photodegradation percentage of RhB over 3% CNT/CuS.

**Figure 12 nanomaterials-10-00178-f012:**
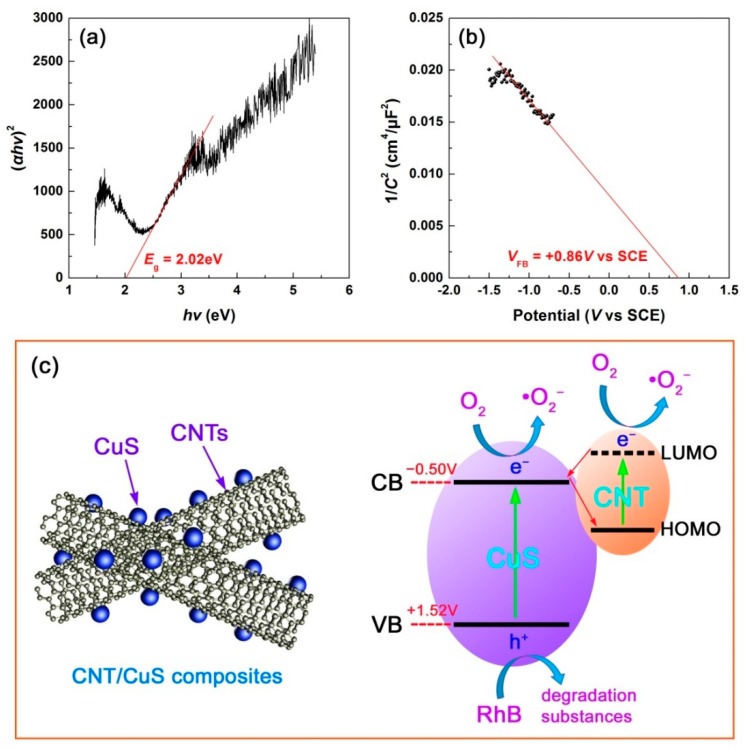
(**a**) Tauc plot of (*αhν*)^2^ vs. *hν* for CuS. (**b**) M-S plot of CuS measured at a given frequency of 5000 Hz. (**c**) Schematic illustration of photodegradation mechanism of the CNT/CuS composite photocatalysts.

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
