# Peer review of "In Situ Construction of CNT/CuS Hybrids and Their Application in Photodegradation for Removing Organic Dyes"

_nanomaterials, 2020, doi:10.3390/nano10010178_

Round 1

Reviewer 1 Report

General comments

The paper presents the results of an experimental study of the material and photocatalytic properties of the CuS/CNT nanocomposites prepared by coprecipitation or simply mechanical mixing, having the CuS photocatalyst as a benchmark, while rhodamine B was the contaminant to be removed from aqueous solution by light irradiation. The results prove the formation of a CuS/CNT nanocomposite by coprecipitation and its   catalytic superiority with respect to CuS.   The manuscript is of good technical value, with a strong experimental basis. However, before being published, some aspects should be improved, as described below.

Specific comments

In the introduction section, the prior art literature concerning the use of CNT/CuS hybrid composites for photocatalytic applications is not cited, if it exists. What is the added value of the manuscript with respect to existing literature of this CuS/CNT nanocomposite ? The evolution of the nanostructure of bare CuS as a function of time and temperature is too simply described. It appears that Ostwald ripening may fit well the transition from nanoparticle to nano-flakes as a function of time at room temperature. At high temperatures, the nanostructure formation for bare CuS may be correlated with the increase of nucleation sites for CuS formation, and this may reduce the chance for anisotropic growth of nano-flakes.  In the case of  CuS/CNT composite formation, the CNT may block the  anisotropic expansion of the crystal seed, and nano-flakes formation. The 3% CuS/CNT-MD composite has the highest catalytic efficiency, and this result cannot be ignored. Actually, it proves the essential role of nano-flake structure in the catalytic process, even if this technology approach was a kind of “mechanical” mixing process.   This result is very important, and it is missing in both abstract and conclusion section. The model of the photocatalytic activity of the CuS/CNT composite has some week points, as follows. The CNT is a good electron acceptor, but for this purpose, its LUMO level should be located below the bottom of the conduction band (CB) of the CuS, and thus there will be no energy barrier seen by those electrons moving towards the CNT. On the other hand there is a good chance for holes to move to the HOMO level of CNT, and this approach will increase the role of CNT in the charge separation of the photogenerated electron-hole pairs . I am attaching  a recent literature for this purpose to you.          

Author Response

Reviewer #1

The paper presents the results of an experimental study of the material and photocatalytic properties of the CuS/CNT nanocomposites prepared by coprecipitation or simply mechanical mixing, having the CuS photocatalyst as a benchmark, while rhodamine B was the contaminant to be removed from aqueous solution by light irradiation. The results prove the formation of a CuS/CNT nanocomposite by coprecipitation and its   catalytic superiority with respect to CuS.   The manuscript is of good technical value, with a strong experimental basis. However, before being published, some aspects should be improved, as described below.

Response: We thank very much the Reviewer for reviewing our manuscript and giving valuable comments and suggestions. According to the Reviewer’s suggestions, we have carefully revised the manuscript.

Specific comments

In the introduction section, the prior art literature concerning the use of CNT/CuS hybrid composites for photocatalytic applications is not cited, if it exists. What is the added value of the manuscript with respect to existing literature of this CuS/CNT nanocomposite ?

Response: The prior art literature concerning the use of CNT/CuS hybrid composites for photocatalytic applications has been cited. Previous work was mainly concerned with the application of CNT/CuS composites in solar cells, semiconductor plasmonics, and supercapacitors. The present work is mainly concerned with the photocatalytic application of the CNT/CuS hybrid composites for the degradation of organic dyes; moreover, in this work we have investigated the evolution process of CuS morphology between nanoparticles and nanoflakes by varying the reaction time and temperature.

The evolution of the nanostructure of bare CuS as a function of time and temperature is too simply described. It appears that Ostwald ripening may fit well the transition from nanoparticle to nano-flakes as a function of time at room temperature. At high temperatures, the nanostructure formation for bare CuS may be correlated with the increase of nucleation sites for CuS formation, and this may reduce the chance for anisotropic growth of nano-flakes.  In the case of  CuS/CNT composite formation, the CNT may block the  anisotropic expansion of the crystal seed, and nano-flakes formation.

Response: We have described in details the evolution of the nanostructure of bare CuS as a function of time and temperature according to the excellent suggestions given by the Reviewer.

The 3% CuS/CNT-MD composite has the highest catalytic efficiency, and this result cannot be ignored. Actually, it proves the essential role of nano-flake structure in the catalytic process, even if this technology approach was a kind of “mechanical” mixing process.   This result is very important, and it is missing in both abstract and conclusion section.

Response: According to the reviewer’s suggestion, we have added the information about the 3% CuS/CNT-MD composite in the Abstract and Conclusion.

The model of the photocatalytic activity of the CuS/CNT composite has some week points, as follows. The CNT is a good electron acceptor, but for this purpose, its LUMO level should be located below the bottom of the conduction band (CB) of the CuS, and thus there will be no energy barrier seen by those electrons moving towards the CNT. On the other hand there is a good chance for holes to move to the HOMO level of CNT, and this approach will increase the role of CNT in the charge separation of the photogenerated electron-hole pairs . I am attaching a recent literature for this purpose to you.

Response: Thank the Reviewer for giving this good comment. We have carefully studied the comment given by the Reviewer. The Reviewer’s view is correct and very useful for type II heterojunction composite photocatalystst. However, the present CuS/CNT composite is different from the type II heterojunction, instead it is somewhat similar to the Z-scheme mechanism. We have provided this information and cited the paper mentioned by the reviewer in the revised manuscript. We hope the reviewer can accepted our proposed mechanism.

Reviewer 2 Report

In the work titled "In-situ construction of CNT/CuS hybrids and their promising photodegradation application for removing organic dyes" the preparation of CuS as catalyst for dyes photodegradation is addressed. The influence of the reaction time in the particles morphology and, therefore, how the catalytic activity is affected. There are some aspects that should be addressed prior to its publication:

How the change in the morphology with the time is explained? Authors should give more details. According to XRD diffractograms there are not any significant difference depending on the different reaction conditions used for the materials, how is explain this apparent contradiction? The addition to the manuscript of a breif explanation about how RhB is degradated together with the reaction scheme would be useful for the reader to understand better what is happening and how CNT/CuS works. Figure 4 shows the differences found on RhB photodegradation for the different CuS materials synthesised at different reaction conditions. Are such differences so relevant according to the materials characterisation? The explanation to the differences on RhB photodegradation (page 6, line 190) "With increasing the synthesis temperature, the derived CuS samples exhibit a decrease in the photodegradation activity. This is because that CuS nanoflakes are evolved into nanoparticles at high reaction temperatures" is very poor.  On line 204, page 6, it is stated that "CuS in the composite is crystallized into 204 nanoparticles, instead of forming nanoflakes". EDS elemental mapping images are used to support that CNTs are uniformly integrated with CuS nanoparticles. Firstly, it is not accurate the use of this technique to evaluate the homogeneity of a sample. Secondly, it is clear just the opossite, on Fig 6h can be clearly seen C agglomerates.  On the FT-IR plot it is obserbed at 630 cm-1 a band corresponding to the Cu-S stretching mode whose intensity decreases in the 3%CNT/CuS spectrum as compared to CuS sample. In addition, the differences between 3%CNT/CuS, CuS and CNT should be more deeply explained. 

Author Response

Reviewer #2

In the work titled "In-situ construction of CNT/CuS hybrids and their promising photodegradation application for removing organic dyes" the preparation of CuS as catalyst for dyes photodegradation is addressed. The influence of the reaction time in the particles morphology and, therefore, how the catalytic activity is affected.

Response: We thank very much the Reviewer for carefully reading our manuscript and giving valuable suggestions. According to the Reviewer’s suggestions, we have carefully revised the manuscript.

There are some aspects that should be addressed prior to its publication:

How the change in the morphology with the time is explained? Authors should give more details.

Response: According the Reviewer’s suggestion, the growth process and mechanism of CuS nanoflakes with the time have been explained in details.

According to XRD diffractograms there are not any significant difference depending on the different reaction conditions used for the materials, how is explain this apparent contradiction?

Response: Thank the reviewer for giving this good comment. The preferential orientation of CuS nanolakes can not be clearly recognized on the XRD diffractograms, a possible reason for which is that the diffraction peaks are partially overlapped each other, as shown in Fig. 3.

The addition to the manuscript of a breif explanation about how RhB is degradated together with the reaction scheme would be useful for the reader to understand better what is happening and how CNT/CuS works.

Response: According to the reviewer’s suggestion, we have briefly explained about how RhB is degradated together with the reaction scheme.

Figure 4 shows the differences found on RhB photodegradation for the different CuS materials synthesised at different reaction conditions. Are such differences so relevant according to the materials characterisation? The explanation to the differences on RhB photodegradation (page 6, line 190) "With increasing the synthesis temperature, the derived CuS samples exhibit a decrease in the photodegradation activity. This is because that CuS nanoflakes are evolved into nanoparticles at high reaction temperatures" is very poor. 

Response: Thank the reviewer for giving this good comment. Much work has shown that the photocatalytic performances of semiconductors are highly dependent on their morphologies. Considering the Rviewer’s comment, we tried our best to modify this part.

On line 204, page 6, it is stated that "CuS in the composite is crystallized into 204 nanoparticles, instead of forming nanoflakes". EDS elemental mapping images are used to support that CNTs are uniformly integrated with CuS nanoparticles. Firstly, it is not accurate the use of this technique to evaluate the homogeneity of a sample. Secondly, it is clear just the opossite, on Fig 6h can be clearly seen C agglomerates. 

Response: We very much agree with the Reviewer’s view, we have made the relative modification, especially the word “uniformly” has been deleted.

On the FT-IR plot it is obserbed at 630 cm-1 a band corresponding to the Cu-S stretching mode whose intensity decreases in the 3%CNT/CuS spectrum as compared to CuS sample. In addition, the differences between 3%CNT/CuS, CuS and CNT should be more deeply explained.

Response: According the reviewer’s suggestion, we have modified this part.

Round 2

Reviewer 2 Report

Authors have done the suggested corrections and answered the questions. The manuscript can be published as it is.